# Real-World Robot Learning with Masked Visual Pre-training

**Ilija Radosavovic*** **Tete Xiao*** **Stephen James** **Pieter Abbeel** **Jitendra Malik**[†] **Trevor Darrell**[†]

University of California, Berkeley

**Abstract:** In this work, we explore self-supervised visual pre-training on images from diverse, in-the-wild videos for real-world robotic tasks. Like prior work, our visual representations are pre-trained via a masked autoencoder (MAE), frozen, and then passed into a learnable control module. Unlike prior work, we show that the pre-trained representations are effective across a range of real-world robotic tasks and embodiments. We find that our encoder consistently outperforms CLIP (up to 75%), supervised ImageNet pre-training (up to 81%), and training from scratch (up to 81%). Finally, we train a 307M parameter vision transformer on a massive collection of 4.5M images from the Internet and egocentric videos, and demonstrate clearly the benefits of scaling visual pre-training for robot learning.

**Keywords:** Self-supervised Learning, Visual Representations, Robot Learning

## 1 Introduction

Learning representations with large neural networks is the workhorse of modern deep learning. This has enabled impressive results in computer vision [1, 2], natural language processing [3, 4, 5], and audio generation [6, 7]. How can we transfer the success stories of representation learning to robotics? We can approach this from two ends: shared representations on the perception side or shared representations on the action side. Our focus in this paper is on shared *visual* representations.

Of course, the devil is in the details. Recent developments in the field of visual learning have made this more feasible: (1) the use of diverse, real-world data from the Internet and egocentric videos, (2) self-supervised objectives that do not overly rely on data augmentations or other forms of strong human-designed priors, (3) scalable and high-capacity transformer models [8, 9], and (4) training of control policies on top of frozen visual representations. In our recent work [10], we have shown that this recipe for self-supervised visual pre-training is effective for motor control in simulation.

In this paper, we show that this framework is effective for real-world robotic tasks as well (Figure 1). We build on our prior work, but make significant advances in terms of data scale and diversity (7× larger), model size (15× bigger), and real-world experiments (extensive real robot evaluations).

In particular, we train *self-supervised* visual representations on real-world images and videos from the Internet [11, 12, 13] and egocentric video datasets [14, 15]. We leverage the masked autoencoders [16] that learn visual representations by masked prediction. The hope is that, by learning to predict the missing content in real-world images, the model will learn useful properties of the visual world that will enable it to learn to perform real-world robotic tasks. Given the pre-trained vision encoder, we *freeze* the encoder and learn control policies on top. The *same* visual representations are used for all downstream robotic tasks and embodiments. We focus on efficient real-world learning through behavior cloning with a handful of human-provided demonstrations per task (20 - 80).

---

*,†Equal contribution. Code, pre-trained models, and videos are available on our project page.

6th Conference on Robot Learning (CoRL 2022), Auckland, New Zealand.

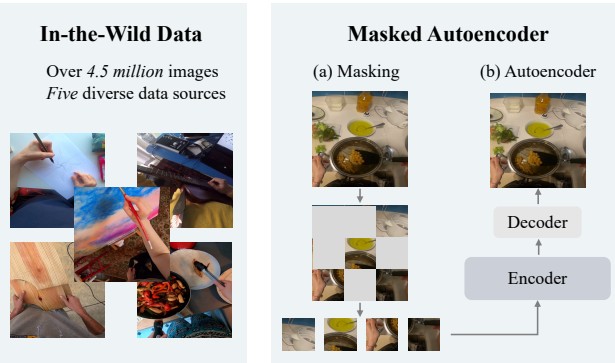
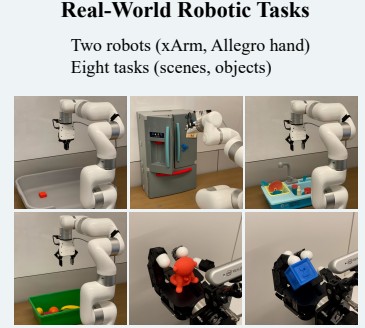

Figure 1: **Real-world robot learning with masked visual pre-training.** We learn visual representations from a massive collection of Internet and egocentric data. We pre-train representations with masked image modeling, freeze the encoder, and learn control policies for robotic tasks on top.

We evaluate our approach in an extensive real-world study and report results from 981 real-world experiments. We consider basic motor control tasks (reach, push, pick), as well as tasks with variations in scenes and objects (Figure 1, right). We find that our approach achieves considerably higher performance than CLIP (up to 75%), supervised pre-training (up to 81%), and training from scratch (up to 81%). Furthermore, we observe that our representations lead to large improvements in sample complexity, reaching the strongest baseline performance with half the number of demonstrations.

In addition, we demonstrate the benefits of scaling visual pre-training for robotics by training a 307M parameter transformer [9] on a massive collection of 4.5M images from ImageNet [11], Epic Kitchens [17], Something Something [12], 100 Days of Hands [13], and Ego4D [15] datasets. Importantly, we observe that it is not sufficient to scale the model alone and that larger models require bigger datasets. To the best of our knowledge, ours is the largest vision model deployed for robotics, and demonstrates clearly the benefits of visual pre-training scale for robot learning. We encourage the readers to see the the extended version of this work on arXiv and also to check the project page.

## 2 Related Work

**End-to-end control** is concerned with learning to predict robot actions (*e.g.*, joint velocities, end-effector poses, etc) directly from observations [18, 19, 20], without the need to perform explicit 3D pose estimation [21], grasp planning [22], and motion planning [23]. However, these end-to-end approaches tend to be too sample inefficient for real-world training. Some works have tried to find a balance between these explicitly pipelined approaches and end-to-end approaches [24, 25, 26].

**Supervised pre-training for robotics** learns one or more pretext tasks through strong supervision and then transfers the representations to downstream robotic tasks. Lin et al. [27] shows that representations learned from semantic tasks such as detection and segmentation correlate with affordance maps for object manipulation. Shridhar et al. [28] use language-supervised CLIP model [29] for learning language-conditioned imitation policy. In concurrent work, Nair et al. [30] explore pre-training visual representations using time contrastive learning and language descriptions from human annotators. These methods all require expert labels or cross-domain supervision.

**Self-supervised learning in robotics** has been explored in a number of settings: learning a dynamic models [31]; learning visual representations from interaction with the environment [32]; learning vision-based grasping policies [33, 34]; learning visual autoencoders [35]; learning spatiotemporal representations through videos [36, 37]; learning visual correspondence [38]; utilizing non-parametric nearest-neighbor retrieval [39]; and conducting visual self-supervised learning on pre-collected demonstrations [40]. These methods require in-domain data collection, and thus may be difficult to extend beyond the training environment and task. In contrast, our approach uses a large-scale and diverse collection of real-world images and videos, making it more generalizable.

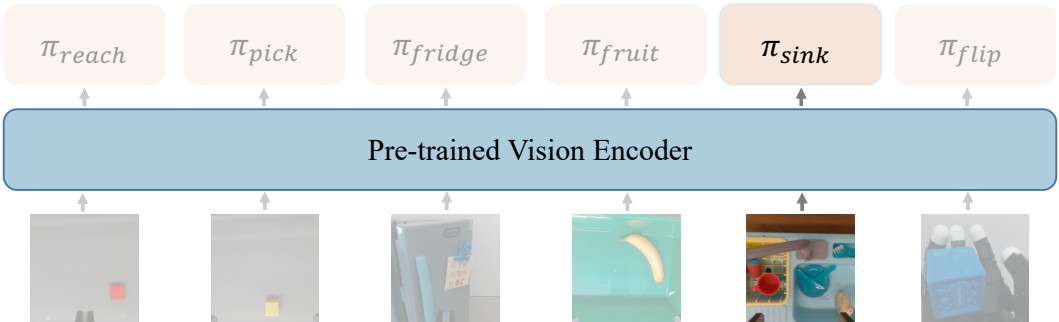

Figure 2: **One encoder for all robots and tasks.** We train control policies per task, on top of the frozen encoder. The same vision encoder is used for all downstream robotic tasks and embodiments.

## 3 Real-World Robot Learning with Masked Visual Pre-training

### 3.1 Masked Visual Pre-training

**Data collection.** We first compile a large-scale dataset for learning visual representations. We primarily use Ego4D [15], a massive scale, egocentric dataset from nine countries recorded via portable devices, covering over 3,670 hours of daily-life activities. We combine the Ego4D data with the ImageNet [11], as well as the Hand-object Interaction (HoI) data used in [10], which comprises of the egocentric Epic Kitchens [17] dataset, the YouTube 100 Days of Hands dataset [13], and the crowd-sourced Something-Something dataset [12]. Our training data totals 4.5 million images, 6.5x of the HoI data. We find that a sufficiently large and diverse pre-training dataset to perform the mask image modeling self-supervisory task is critical to scale up the vision backbone for real robot tasks.

**Self-supervised objective.** At the core of our self-supervised representation learning approach is masked image modeling via the masked autoencoders (MAE) [16]. MAE masks out random patches in an image and reconstructs the missing content with a vision transformer (ViT) [9]. A high masking ratio, *e.g.*, 75%, and asymmetrical heavy-encoder light-decoder design, are important for learning good visual representations efficiently. Simple and free from dataset or task-specific augmentations [41], MAE is the state-of-the-art self-supervised framework in computer vision [42, 43, 44, 45], and has been demonstrated to work well for motor control tasks in simulation as well [10].

**Architecture.** We use the ViT models as our vision encoders. While the MAE-trained ViT models yield improving performance in vision tasks as model sizes grow [9, 16, 46], previous work [10] does not show improvement from switching a ViT-Small model to the ViT-Base counterpart of 4x as many parameters. In this work, we scale the model up to the ViT-Large and deploy it on the real robot. The model contains 307M parameters and runs at ~64 gigaflops at input size 224×224, approximately 15x as many as the commonly adopted ResNet-50 [47], the largest vision model deployed for robotics. As we will show in the experiments, scaling model sizes while training on sufficiently large data leads to consistent performance improvement on downstream robotic tasks.

### 3.2 Real-World Robot Learning

We learn to perform real-robot tasks through behavior cloning (BC). We collect demonstrations containing trajectories of RGB images from a wrist-mounted camera and the robot's joint state at each time step. For most of the tasks, we use the motion-tracked HTC Vive VR system to control the end-effector. For some tasks that are difficult to demonstrate via the motion controller, *e.g.*, closing fridge door, we use kinesthetic teaching. Given the recorded demonstrations, we train a control policy that takes in the input image features and proprioceptive states (joint positions) at time step $t$ and predicts the action at time step $t + 1$. We perform joint position control; we do not use any end-effector information (*e.g.*, the 6-DoF pose). We build on our MVP pipeline [10] and freeze the image encoder throughout the policy learning, which prevents large pre-trained encoders from overfitting to a specific setting or task, and greatly reduces GPU memory footprint and training time.

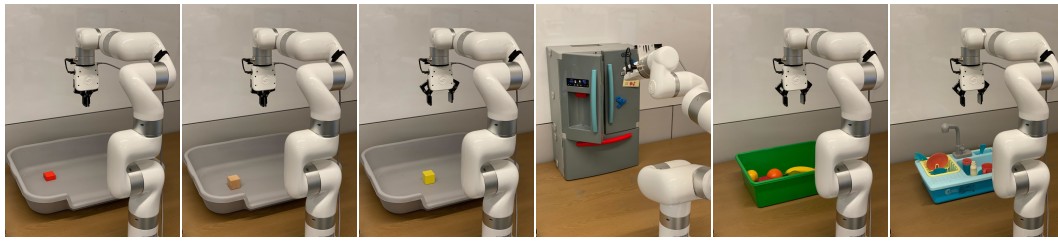

Figure 3: **Real-world robotic tasks.** We perform extensive real robot evaluations using a 7 DoF robot arm with a parallel jaw gripper. Our tasks include basic motor control skills (reaching a red block, pushing a wooden cube, and picking a yellow cube), variations in scenes (closing a fridge), objects (picking fruits), and scenes and objects (picking a detergent bottle from a cluttered sink).

## 4    Experimental Setup

In this section we provide implementation details for our approach and describe our evaluation setup.

**Data.** We extract frames from Ego4D, Epic Kitchens, and Something-Something at 0.2 fps, 1fps and 0.3fps, respectively. We then combine the Ego4d with ImageNet and the YouTube 100 Days of Hands dataset. This process yields 2.6M frames from Ego4D, 1.2M images from ImageNet, and 700k HoI images from the rest, a total of 4.5M images. We term the combined dataset as "Ego" for abbreviation. Note that [10] only uses the 700k HoI images, excluding the Ego4D and ImageNet.

**Encoders.** We use the standard Vision Transformer (ViT) architecture as the image encoder. We use three models of various sizes: ViT-Small [48], ViT-Base, and ViT-Large models, of 22M, 86M, and 307M parameters, respectively. The ViT-Small model is approximately the same size as the ResNet-50 model, while the ViT-Large model has ~15x as many parameters as the ResNet-50 model.

**Pre-training.** We pre-train the models via the MAE framework [16]. The training recipe closely follows [16], with dataset specific settings from [10]. We use the auxiliary dummy classification token for transferring to downstream robotics tasks. We train the MAE models for 400 epochs for the combined Ego dataset; 1600 epochs for the HOI dataset; and 1600 epochs for ImageNet dataset. We use the pre-training recipe in [49] for the study that involves ImageNet supervised models.

**Controllers.** The controller takes in both image features and the robot's proprioceptive state. We use joint positions as the proprioceptive state *without* explicitly appending the end-effector pose to the state. We do not use velocity in the state as our low-cost arm does not support true velocity sensing. The controller outputs *delta joint angles*. The controller's design closely follows [50], *i.e.*, a four-layer MLP with a SeLU [51] activation following each hidden layer. The hidden size is [256, 128, 64] for most tasks and [512, 256, 128] for the `PickSink` task. We linearly project the image features and the proprioceptive states to a joint embedding space as the controller's input.

**Robot and robotic tasks.** We use the low-cost UFACTORY xArm7 robot (a 7-DoF arm) and a 1-DoF parallel jaw gripper. We use the arm's maximum control frequency of 5 Hz for both collecting demonstrations and control. We use a first-person wrist-mounted RealSense camera for all tasks. We do *not* use depth information from the camera. We consider basic motor control tasks, *i.e.*, `ReachBlock`, `PushCube`, and `PickCube`, and more challenging in-context tasks, *i.e.*, `CloseFridge`, `PickFruit`, and `PickSink`. The tasks are shown in Figure 3 (more details in the next section).

**Demonstrations.** We collect 80 demonstrations per task. We use the motion-tracked HTC Vive VR system for most tasks, except for `CloseFridge` we use kinematics teaching. We use trajectory replay on the robot for trajectory pruning. We do *not* use the key-frame information for the learning.

**Evaluation.** We systematically sweep across 16 variations of the environment, *e.g.*, shifting the target object. For consistency and reliability of the study, we use the *same* 16 variations for all models in an individual task, and evaluate models *sequentially* at each variation, in order for similar lighting conditions, robot conditions, precise object initial locations, etc (see also the arXiv).

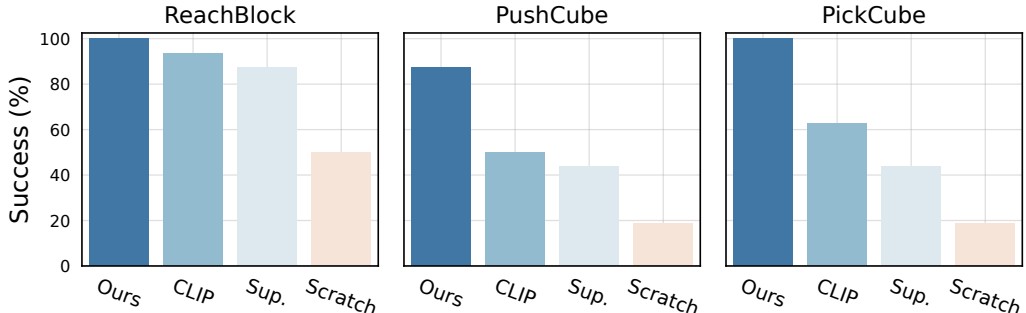

Figure 4: **Comparison to vision encoders.** We compare our approach to visual encoders trained with CLIP, supervised learning on the ImageNet, and from scratch on the task at hand. In all cases, we observe that our approach consistently outperforms the baselines by a considerable margin.

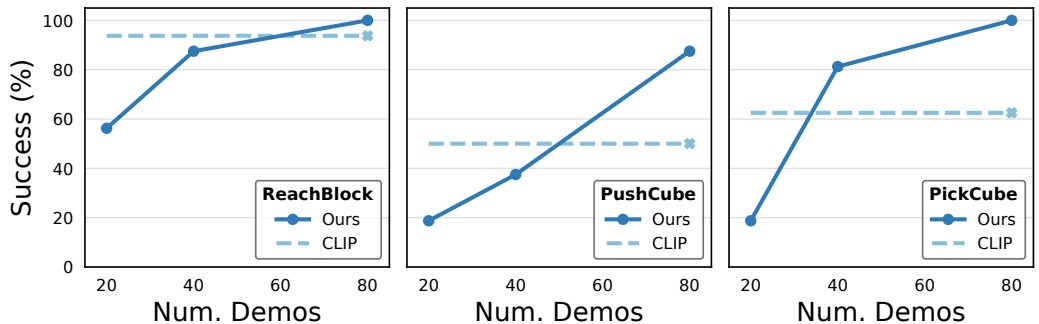

Figure 5: **Sample complexity.** We show the performance of our approach as the number of demonstrations varies from 20 to 80. CLIP performance at 80 demonstrations is shown with a dashed lined for reference. Our approach is comparable to CLIP using only half the number of demonstrations.

## 5 Experimental Results

We perform extensive evaluations across a range of visual backbones, real-world robotic tasks, objects, and environments (Figure 3). In total, we report results from 981 real-world experiments.

### 5.1 Basic Motor Control

We evaluate three basic motor control tasks in visually simple contexts: reaching a red block, pushing a wooden cube, and picking a yellow cube (see Figure 3 for task visualization). These tasks serve as stepping stones for more complex tasks, and the visual representations that can potentially be fundamental for robotics should learn these tasks efficiently. In all cases, we randomize the initial object and robot positions (see arXiv for details about the learning and task setup).

**Comparison to various vision encoders.** In Figure 4 we compare our approach to a set of state-of-the-art vision backbones: CLIP [3] trained on 400M text-image pairs, supervised model trained on the ImageNet, and a model trained from scratch with in-domain demonstration data. For fair comparisons, we use the ViT-Base [9] vision encoder for all methods. We empirically observe that the CLIP encoder performs the best among the baselines, and the ranking order is consistent across the benchmark tasks. Our approach consistently outperforms the baselines by a considerable margin.

**Sample complexity.** In Figure 5 we study the performance of our approach as the number of demonstrations varies from 20 to 80. For reference, we show the performance of the most competitive baseline, CLIP, trained with 80 demonstrations (dashed horizontal line). In aggregate, we observe that our approach reaches CLIP performance while using 50% fewer demonstrations. This result is a promising signal for using our models for learning more complex robotic tasks, as discussed next.

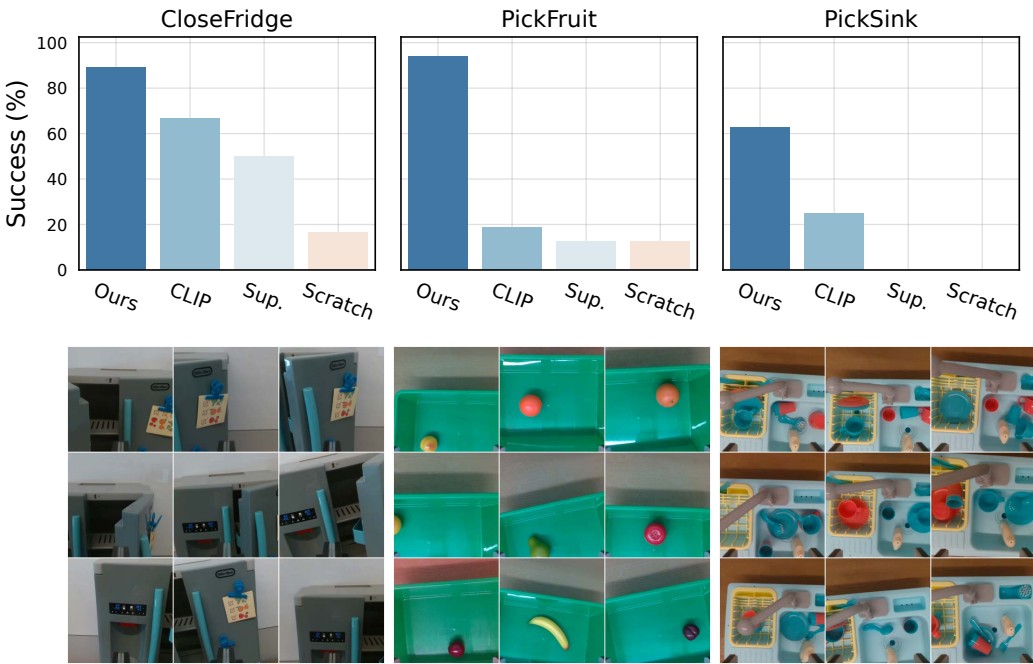

Figure 6: **Variations in scenes and objects.** We compare our approach to CLIP on tasks with variations in scenes (closing the fridge), objects (picking fruits), and scenes and objects (picking an object from a cluttered sink). The models are ViT-Base. Our approach considerably outperforms CLIP and the gap is larger than in simpler settings (see Figure 4). This may suggest that our representations capture more precise spatial structure that is helpful for robotic tasks in more realistic contexts.

## 5.2 Visually Diverse Scenes and Objects

We have previously shown that our approach can learn basic motor control tasks, such as reaching, pushing, and picking, at a higher success rate and better sample complexity than the baseline vision backbones. To focus on the motor control aspect, we used a visually simple environment and basic objects. However, one of the main potential benefits of our approach is that learning visual representations from diverse, real-world images may enable to solve robotic tasks that involve the interaction with everyday objects in more visually complex environments. In this subsection, we evaluate our visual representations on robotic tasks with variations in scenes and objects in more realistic setups.

**Scene context.** We first evaluate our approach in a more realistic scene by considering the task of closing the door of a toy fridge that is left open (termed `CloseFridge`). We randomize the location of the fridge, which side of the door to close, and initial angle of the door. As shown in Figure 6, bottom-left, the initial configuration of the fridge and the robot vary considerably, which is quite common in everyday settings. Figure 6, top-left, shows that our approach outperforms all baselines.

**Different objects.** Next, we evaluate on a task with a variety of objects. The goal is to pick up eight different fruits that vary in color, shape, and size (termed `PickFruit`). In each trial, a fruit is selected at random, and both the fruit and the robot's positions are randomized. The number of training demonstrations remain unchanged, *i.e.*, we provide 10 demonstrations for each fruit. In Figure 6, middle, we show the evaluation results (top) and starting configuration samples (bottom). We see that baselines struggles in this setting while our approach achieves nearly perfect score.

**Objects in context.** Finally, we evaluate our approach on a task that features interacting with objects in everyday contexts. We task the robot with picking a detergent bottle from a cluttered sink (termed `PickSink`). The task is challenging as the visual configuration of the scene, such as the toy plates, mug cups, and silverware, can vary in unlimited ways, as shown in Figure 6 right We observe that our approach considerably outperforms baseline approaches using the same ViT-B encoder.

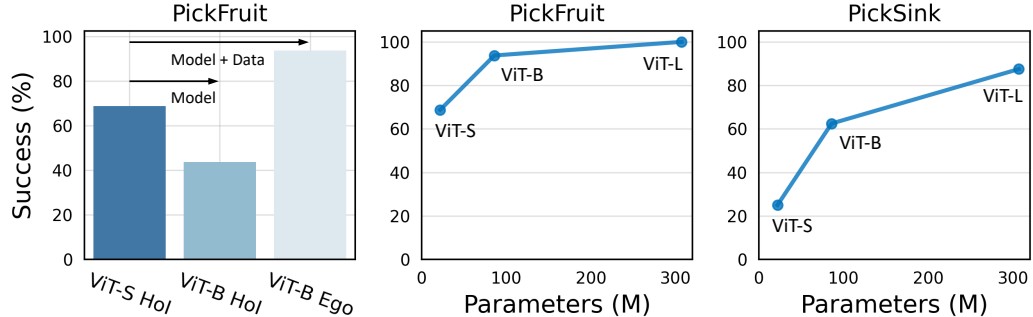

Figure 7: **Scaling model and data.** We study the scaling properties of our approach. We observe that scaling the model size alone from ViT-S to ViT-B while keeping the dataset fixed (HoI image collection; see text for details) does not improve the performance and even hurts (left). However, when we scale both the model and data (our massive Ego image collection; see text for details) we see clear benefits from a larger model. The trend continues when going further from the 86M ViT-B to the 307M ViT-L model (middle & right). Moreover, the gains are larger for harder tasks (right).

## 5.3 Scaling Model and Data Size

Importantly, our visual pre-training approach uses a self-supervised objective [16] that makes few assumptions about the data distribution, and does not rely on human-designed pretext tasks such as data augmentations. Therefore, the framework is well-suited for pre-training from a massive collection of unlabeled and in-the-wild visual data. Here we study scaling model and data size.

We first consider increasing the model capacity. In Figure 7, left, we see that increasing the model size (~4.5x) from ViT-S to ViT-B, while keeping the data size fixed (HOI image collection [10]), does not increase performance and even hurts. This is consistent with the in-simulation results reported in [10]. However, if we also scale the data size from HOI to our massive Ego data collection, ViT-B yields better results. These results suggests that we must scale *both* the model and the data.

In Figure 7, middle & right, we show the performance as a function of model size. Additionally increasing the model size from the 86M parameter ViT-B to the 307M parameter ViT-L leads to further improvements. The gain is larger for the visually more challenging task (`PickSink`). To the best of our knowledge, our work is the largest vision model deployed to real robot tasks, which clearly demonstrates the benefits of scaling visual pre-training for real-world robot learning.

## 5.4 Comparison to Concurrent Work

We compare our approach to a concurrent work [30], submitted to the same conference (CoRL 2022). Similarly, it pre-trains visual representations on in-the-wild video data from Ego4D [15]. However, it relies on paired language-video annotations that are available as part of Ego4D. In contrast, our approach is fully self-supervised and makes minimal assumptions about the data distribution, enabling us to leverage massive collections of uncurated data (*e.g.*, from the Internet). In Table 1, we compare our models to the strongest available R3M ResNet-50 model. We observe that our medium-sized ViT-B model outperforms R3M ResNet-50 by a large margin (93.8% *vs.* 31.3%). We also see that our smallest ViT-S model from [10] outperforms it as well (68.8% *vs.* 31.3%).

|  | supervision | params (M) | success (%) |
|---|---|---|---|
| R3M | video-text | 23 | 31.3 |
| CLIP | image-text | 86 | 18.8 |
|  |  | 22 | 68.8 |
| Ours | image-only | 86 | 93.8 |
|  |  | 307 | 100.0 |

Table 1: **Comparison to concurrent work.** All of our vision models, trained with image-only self-supervision, considerably outperform the strongest available R3M [30] model trained on paired video-language labels from Ego4D [15]. The gains are larger for larger models. Evaluated on the PickFruit task.

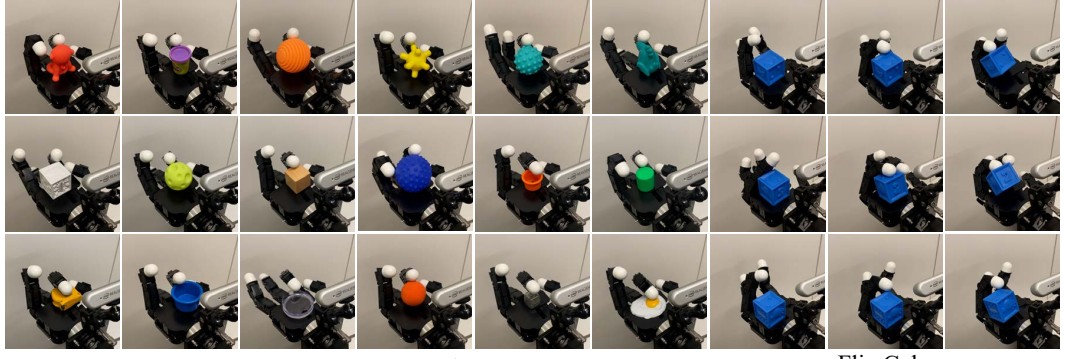

Reach Seen                    Reach Unseen                        Flip Cube

Figure 8: **Multi-finger hand.** We show that our framework readily generalizes to a different robot morphology. We experiment with finger reaching, using seen and unseen objects, and cube flipping.

### 5.5 Case Study: Multi-finger Hand

Our approach makes no assumptions about the downstream robotic tasks or embodiments. In particular, our policies take pixel images as input and predict joint position angles as actions (rather than, *e.g.*, end-effector pose). In this subsection, we test the generality of our approach by applying it for downstream tasks with a multi-finger Allegro hand (see arXiv for more details on the setup).

**Visual reaching.** We design a reaching task in which the hand learns to reach the top of an object with the tip of the index finger. The object's position is randomized across the palm. We provide 10 demonstrations for each of the 8 different objects. At test time, we evaluate the trained policy on the 8 seen objects as well as 45 unseen objects (see Figure 8). The success rate is ~50% across both.

**Visual flipping.** Next, we consider a cube flipping task in which the goal is to flip a rubber cube that is placed in the palm. The position of the cube is randomized across the palm. Thus, the policy must rely on visual cues to accomplish the task. In aggregate, we observe a success rate of 50% across 30 trials. See Figure 8 for example key frames and also check out the videos on the project page.

**Understanding vision through action.** Training policies on top of frozen visual representations enables us to perform studies to understand what the pre-trained visual representations utilize for downstream tasks. Here we first train a policy to reach a yellow cube, but test with objects of different shape and color. First, we find that when given the same shape of different color (wooden cube) or same color and different shape (yellow ball), the policy reaches for the object. Next, when given both the yellow cube and a distractor it reaches for the yellow cube. Finally, when given an object of different shape and color (blue cube) the hand stays still. See project page for videos.

## 6 Discussion

**Limitations.** While the scenes and objects used in our study are more realistic than in simpler robotic benchmarks, we still mostly use toy objects in relatively clean lab environments, rather than real objects in real-world scene contexts. Our tasks involve a single object, and do not require the robot to learn feedback control, or to learn multi-step planning. Factors such as robot up-time may influence the results as well. Overcoming these limitations is essential as we move toward developing, benchmarking, and widely deploying pre-trained models for real-world robotic applications.

**Conclusion.** We explore learning visual representations from a massive collection of real-world data and using them for downstream robotic tasks. We pre-train representations with masked modeling, freeze the encoder, and learn control policies on top. We perform extensive evaluations in the real world and show that, across various robotic tasks, our approach leads to higher success rate and better sample complexity than CLIP, supervised ImageNet pre-training, and training from scratch. We further demonstrate the benefits of scaling the model and data size for real world robot learning.

**Acknowledgments**

We gratefully acknowledge the following colleagues for valuable discussions and support of our project: Shankar Sastry for Allegro hand access, Ken Goldberg for feedback and suggestions, Adam Curtis for help with wiring, Kevin Hu for help with Allegro hand retargeting, Erik Rogers for ROS suggestions, Lerrel Pinto for Allegro hand discussions, Raven Huang and Justin Kerr for help with 3D printing, and William Peebles for discussions. This work was supported in part by DARPA Machine Common Sense, LwLL, and RACER programs; ONR MURI program (N00014-21-1-2801); Hong Kong Centre for Logistics Robotics, BMW, as well as BAIR's industrial alliance programs.

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
