# OpenReview forum: "Real-World Robot Learning with Masked Visual Pre-training"
_robot-learning.org/CoRL/2022/Conference — CoRL 2022 Oral_

### Official Review · Reviewer_ZgtC · 2022-07-29

**Originality:** Fair
**Technical Quality:** Very Good
**Clarity Of Presentation:** Very Good
**Impact:** 3

**Recommendation:**

Weak Accept: I recommend accepting the paper, but will not argue for my recommendation if the majority of other reviewers have a different opinion.

**Summary:**

This paper scales up vision models and data in order to train better vision backbones for visuomotor policies using behavior cloning. They aggregate several existing datasets in order to generate a large number of frames to pretrain a large high quality vision encoder (VIT using MAE). The authors conduct extensive experiments to show that scaling both the model and data substantially improves performancce on these tasks.

**Issues:**

I don't have any major issues. Minor complaints: I would replace "demos" with "demonstrations" everywhere. The graphs could use better color choices with more contrast.

**Quality Of The Limitations Section:**

Limitations section not present

**Reviewer Expertise:**

4: The reviewer is confident but not absolutely certain that the evaluation is correct

**Robotics Focus:**

Sufficient demonstration on hardware

**Strengths And Weaknesses:**

Stengths:
The paper is well presented and makes its points clealy.
The authors conduct many real world experiments to demonstrate the effectiveness of their approach.
Their new model performs very well against reasonable baselines.

Weaknesses:
The only real weakness here is that there is very little technical novelty. At this point everybody knows scaling up to larger models with more data is a good way to go.

More Strengths:
Despite the weakness above, this paper is still a valuable contribution to the community. While the large ideas here are all well known, the specific recipes are useful and will provide others with practical technical guidance.
It is also useful to see that so much of the performance deficit on these problems can be solved by simply improving the vision model.

**Summary Of Recommendation:**

The lack of technical novelty prevents me from giving this a strong accept, however this paper is still a very useful resource for practioners and those trying to improve robotic capabilities. For "Impact" I wish there was another option: "3.5: The work is incremental and does not contain many new interesting ideas, but will still have significant impact due to its strong experiments and useful practical guidance." Ultimately though, this paper passes an important test: if I were starting a new project in this area tomorrow, this is a paper I would want to read beforehand.

---

> ### Author Response · Authors · 2022-08-28
> **Authors' Response**
>
> Thank you for your comments and suggestions. Please find our comments below:
>
> > The lack of technical novelty prevents me from giving this a strong accept […] but will still have significant impact due to its strong experiments
>
> We believe it is ok to give a strong accept based on the novelty of the empirical results per the [review criteria](https://corl2022.org/instructions-for-authors/) from the CoRL 2022 website: “Submissions will be evaluated based on the significance and *novelty of the results*, either theoretical or *empirical*.”
>
> > Ultimately though, this paper passes an important test: if I were starting a new project in this area tomorrow, this is a paper I would want to read beforehand.
>
> We really like this criteria and thank you for articulating it.
>
> > I don't have any major issues. Minor complaints: I would replace "demos" with "demonstrations" everywhere. The graphs could use better color choices with more contrast.
>
> We will update the manuscript per your suggestions.

---

### Official Review · Reviewer_tTKM · 2022-08-01

**Originality:** Excellent
**Technical Quality:** Excellent
**Clarity Of Presentation:** Excellent
**Impact:** 4

**Recommendation:**

Strong Accept: I recommend accepting the paper and will argue for my recommendation even if other reviewers hold a different opinion.

**Summary:**

This paper shows a way to use large scale visual self supervision of in-the-wild, massive datasets, and use it to improve behavior cloning on robots. Specifically they use the Masked Autoencoder (MAE) framework of He et al, on a massive dataset of {ImageNet, Ego4D, etc.} image frames, to pretrain an encoder which shows improvements for training in-hand-camera-based BC policies.

**Issues:**

See weaknesses discussion.

**Quality Of The Limitations Section:**

Limitations are addressed clearly

**Reviewer Expertise:**

5: The reviewer is absolutely certain that the evaluation is correct and very familiar with the relevant literature

**Robotics Focus:**

Sufficient demonstration on hardware

**Strengths And Weaknesses:**

# Strengths

Strong accept.

- While considerable work has investigated visual pre training to improve robot policy learning and for example behavior cloning, the key thing about this work compared to prior works is that it can benefit from pre training on just a big pile of images, with no assumptions about them other than that they cover a wide, diverse set of images.
- It is indeed also notable that they show improvements of using ViT-Large, a 307M parameter model. For those that believe in the “just scale things up” hypothesis of robotics, this indicates a potentially very compelling route forward.
- The experiments seem pretty good too. I think all the experiments only work for in-hand cameras, and all the tasks are basically just some form of a “reaching” task without any dexterous closed loop feedback required, but they are still good enough to interestingly explore some difficulties in this domain, such as the cluttered “sink” environment.
- I appreciate the different types of scaling model size and/or data size experiments.
- Although [9] is closely related and probably from the same set of authors, the present work is very interestingly different. For one, [9] used RL, and required millions of environment interactions, and was only shown in simulation. Instead, the present work shows real robot results and instead uses behavior cloning. It would be interesting to see RL on the real robots too, but of course it’s easier to deploy BC instead of on policy RL in practice. Also, the present work shows further scaling of vision model size than was shown in [9].

# Weaknesses

To further improve the paper, the authors may consider addressing:

- There are no videos of the Allegro experiments. It’s hard to evaluate these without videos. Currently they feel like kind of throwaway experiments.
- How was imitation learning actually done? I assume it’s just M.S.E loss behavior cloning, but it’s not actually specified?
- As mentioned above the tasks are interesting in some ways (clutter) but pretty simplistic in other ways. No feedback or multi-object-conditioning is required. Harder tasks would be interesting in future.
- The discussion for related work could further be improved. Within self supervised vision for robotics, there have been many approaches that have shown improvements for policy learning. See for example https://arxiv.org/pdf/1909.06933.pdf. But I think the key thing is that the present paper shows using a significantly more massive dataset for pre training and with no assumptions about the data.
- “In recent work, Xiao et al. [9] have shown that self-supervised visual pre-training is effective for learning motor control tasks in simulation.” Respectfully there is nothing in this statement that is unique to Xiao et al, it would probably be better to either more specifically call out the MAE method used in Xiao et al, and/or refer instead to the concept that it used self supervision on internet data, or instead make this a more generic statement about the field rather than specific to Xiao et al.

**Summary Of Recommendation:**

Great strengths outweighs minor weaknesses. Strong accept. Largest scale of vision data in vision-for-robots work I know of, and compelling results.

---

> ### Author Response · Authors · 2022-08-28
> **Authors' Response**
>
> Thank you for your comments and suggestions. Please find our comments below:
>
> > There are no videos of the Allegro experiments. It’s hard to evaluate these without videos. Currently they feel like kind of throwaway experiments.
>
> Please find the videos of the Allegro Hand study in the new website attached in the [general response](https://openreview.net/forum?id=KWCZfuqshd&noteId=Wll_sAdo536) (at the end of the website).
>
> > How was imitation learning actually done? I assume it’s just M.S.E loss behavior cloning, but it’s not actually specified?
>
> That is correct: we use BC with delta joint position actions and MSE loss. We will include in the text and release the code used for the experiments.
>
> > As mentioned above the tasks are interesting in some ways (clutter) but pretty simplistic in other ways. No feedback or multi-object-conditioning is required. Harder tasks would be interesting in future.
>
> We agree and will include in the discussion of limitations.
>
> > The discussion for related work could further be improved. Within self supervised vision for robotics, there have been many approaches that have shown improvements for policy learning. See for example https://arxiv.org/pdf/1909.06933.pdf. But I think the key thing is that the present paper shows using a significantly more massive dataset for pre training and with no assumptions about the data.
>
> Thank you for the reference; we will include (we are happy to include any additional references as well). We also agree about the key aspect and will clarify in the improved discussion of related work.
>
> > In recent work, Xiao et al. [9] have shown that self-supervised visual pre-training is effective for learning motor control tasks in simulation.” Respectfully there is nothing in this statement that is unique to Xiao et al, it would probably be better to either more specifically call out the MAE method used in Xiao et al, and/or refer instead to the concept that it used self supervision on internet data, or instead make this a more generic statement about the field rather than specific to Xiao et al.
>
> We think that this is a fair comment and will revise the text per your suggestion.

---

### Official Review · Reviewer_yV1f · 2022-08-01

**Originality:** Very Good
**Technical Quality:** Very Good
**Clarity Of Presentation:** Very Good
**Impact:** 4

**Recommendation:**

Strong Accept: I recommend accepting the paper and will argue for my recommendation even if other reviewers hold a different opinion.

**Summary:**

This paper evaluates vision encoders pretrained using masked autoencoders on large scale datasets. Pretrained vision encoders are evaluated for real world robotics tasks, with variations of everyday scenes, as well as novel objects.

**Issues:**

* Current results only show graphs, however tables showing more precise numbers would be easier to read
* Authors should consider adding more comparisons/ablations as mentioned in weaknesses above.

**Quality Of The Limitations Section:**

Limitations are addressed clearly

**Reviewer Expertise:**

3: The reviewer is fairly confident that the evaluation is correct

**Robotics Focus:**

Sufficient demonstration on hardware

**Strengths And Weaknesses:**

Strengths
* Evaluating performance for real world policies by pretraining on scalable datasets is an important problem that this paper addresses.
* The real world experiments are good and contain some variety of tasks, demonstrating that pretraining using masked autoencoders works better than a CLIP baseline for several tasks.
* Ablations comparing different camera viewpoints, model and data size.

Weaknesses
* The baselines compare to only ViT architectures. However these are generally harder to train than simpler conv architectures like ResNet. I would be interested in comparing to a from-scratch resnet50 (or smaller) which is easier to train, as another comparison point. Authors could also compare to the ResNet CLIP model. Comparing to visual encoders that are finetuned would also be valuable, since simpler architectures may improve with finetuning when training the policies on demonstration data.
* Other comparisons and ablations would be useful for comparing effectiveness of method - for example: 1) evaluating the effectiveness fine-tuning the vision encoder during the policy learning phase,  2) comparing directly to R3M loss formulation [Nair et. al.]
* Experiments in sim could allow more comprehensive and scalable evaluations.



**Summary Of Recommendation:**

Overall the paper is clear and demonstrates an interesting result for an important problem - that using masked autoencoders to learn visual representations from scalable datasets is effective for robotics policies in the real world.

---

> ### Author Response · Authors · 2022-08-28
> **Authors' Response**
>
> Thank you for your comments and suggestions. We performed all of the additional comparisons suggested in the review. Please find the results and comments below (also included in the new website attached in the [general response](https://openreview.net/forum?id=KWCZfuqshd&noteId=Wll_sAdo536), for convenience):
>
> > The baselines compare to only ViT architectures. However these are generally harder to train than simpler conv architectures like ResNet. I would be interested in comparing to a from-scratch resnet50 (or smaller) which is easier to train, as another comparison point.
>
> Results figure: [link](https://imgur.com/a/QGPmEr6). We provide new results of training ResNet-18 and ResNet-50 models from scratch, and include results of ours and from-scratch ViT-B from the original manuscript for reference (s indicates scratch). We see that when training from scratch, the larger ResNet-50 outperforms the smaller ResNet-18, and is on par with the ViT-B model. All from-scratch models significantly underperform our method.
>
> > Authors could also compare to the ResNet CLIP model.
>
> Results figure: [link](https://imgur.com/a/8IgKz6n). We provide new results of the CLIP ResNet-50 model, and include the results of ours and CLIP ViT model from the original manuscript for reference. We see that CLIP ResNet-50 model underperforms CLIP ViT-B model, which aligns with the ranking order of the two CLIP models on vision tasks.
>
> > Comparing to visual encoders that are finetuned would also be valuable, since simpler architectures may improve with finetuning when training the policies on demonstration data.
>
> Results figure: [link](https://imgur.com/a/CHgCYcr). We provide new results of finetuning two sets of models, i.e., ViT-B and ResNet-50, that have room to improve on the task. We observe both sets of models improve marginally. We note that the tradeoff for the small performance gain is requiring one vision encoder for each individual task due to fine-tuning.
>
> > Comparing directly to R3M loss formulation [Nair et. al.]
>
> Results figure: [link](https://imgur.com/a/AfnIz3H). We provide new results of evaluating the open-sourced R3M representations (R-50; the strongest model provided by the authors), and include the results of ours from the original manuscript for reference. We observe that our method significantly outperforms R3M.
>
> > Experiments in sim could allow more comprehensive and scalable evaluations.
>
> Results figure: [link](https://imgur.com/a/DXYh4FE). We provide new results of comparing all vision encoders on a simulated pick cube task. We use the open-sourced environment setup from [Xiao et al., 2022] for the simulated task. We first train an expert policy, and use the policy to collect demos for behavior cloning. We observe the same ranking order at 20, 40 and 80 demos as the PickCube task on a real robot.
>
> *We hope that the additional experiments address your concerns and politely ask you to consider updating your score based on the additional experiments.*

---

### Official Review · Reviewer_oQo2 · 2022-08-03

**Originality:** Good
**Technical Quality:** Good
**Clarity Of Presentation:** Very Good
**Impact:** 4

**Recommendation:**

Strong Accept: I recommend accepting the paper and will argue for my recommendation even if other reviewers hold a different opinion.

**Summary:**

- This paper leverages the recent breakthrough of Masked AutoEncoders (MAE) for the domain of robotics.
- It shows that visual knowledge can be transferred from different domains such as ego4d into specific robotics domains thanks to MAE pretraining.
- The paper finds that MAE is superior to CLIP pretraining.
- It provides some analysis of data and model scales and finds that large data and large model works the best.
- This is evaluated on 4 real but toy scenes with picking, reaching and pushing skills, with high accuracy results for the MAE pretraining.


**Issues:**

- while the scenes are more realistic than some simpler robotics benchmarks, calling them “realistic” is also a little bit of a stretch, the scenes are not every day scenes with real objects, these are toy objects. see comments above.

**Quality Of The Limitations Section:**

Limitations are addressed clearly

**Reviewer Expertise:**

5: The reviewer is absolutely certain that the evaluation is correct and very familiar with the relevant literature

**Robotics Focus:**

Sufficient demonstration on hardware

**Strengths And Weaknesses:**

strengths
- good to see that somewhat different domain data such as ego4d can be useful for this robotics setup
- scaling to large models with large data improves performance, similar to trends seen in foundation models in other domains, this is a good because it means it can leverage large datasets from non robotics sources.
- the advantages of MAE are kept here: minimal assumptions about data and not relying on specific data augmentation
- the analysis of data and model sizes is useful information for the robotics domain (as opposed to just the vision domain from the MAE paper)
- approach generalizes to different morphologies without finetuning

weaknesses
- no quantitative results or videos for hand experiment
- not yet fully real scenes, setups are still somewhat simplistic with uniform backgrounds, sink and fridge scenes are a bit more complex but -not yet real:
  - for the fridge door, the arm starts quite close to the door and only has to decide if it pushes the left side or the right side.
  - for bucket scenes, objects are always at the same ground distance relative to the robot, this might be the reason why depth was not required to solve these tasks, but this will be different in real scenes where the robot body can be at any distance to the objects.
- a pretty direct application of MAE, while it’s great to see it in a robotics setting, the innovation is pretty incremental


**Summary Of Recommendation:**

A decent paper, the robotics experiments are somewhat simple and limited compared to a truly realistic domain, a decent analysis and description of the model. It's somewhat of an incremental paper given that it's an application of recent breakthrough in self-supervision vision with MAE, however bringing that breakthrough to the domain of robotics might be a pivotal moment where robotics finally is able to leverage massive vision datasets. This paper is one example of that being possible, for that I think it deserves an accept.

---

> ### Author Response · Authors · 2022-08-28
> **Authors' Response**
>
> Thank you for your comments and suggestions. Please find our response below:
>
> > while the scenes are more realistic than some simpler robotics benchmarks, calling them “realistic” is also a little bit of a stretch, the scenes are not every day scenes with real objects, these are toy objects. see comments above.
>
> We think that this is a fair comment and we agree with it. We will revise the manuscript to clarify and include in the limitations section.
>
> > no quantitative results or videos for hand experiment
>
> Please find the videos of the Allegro Hand study in the new website attached in the [general response](https://openreview.net/forum?id=KWCZfuqshd&noteId=Wll_sAdo536) (at the end of the website).

---

### Author Response · Authors · 2022-08-28
**Summary of Authors' Response**

**Comment:**

We sincerely thank the reviewers for their time and effort spent on providing careful reviews. We performed all of the additional experiments suggested by Reviewer yV1f, posted comments to each of the reviewers individually, and will update the manuscript accordingly. Please see the attachment for a website with the additional experiments and videos.

**Zip File:**

/attachment/c05016afd2acc118a857cbdd881e7fd0b4525743.zip

---

### Meta-Review · Area_Chair_kYhN · 2022-08-06

**Recommendation:** Accept (Oral)
**Confidence:** 4

**Metareview:**

This paper uses a masked autoencoder as visual pretraining for robot manipulation tasks. The paper shows strong results and received positive initial reviews.  The additional experiments provided during the rebuttal strengthen the paper's contribution claim.

**Best Paper Nomination:**

No